# Phytochemistry, Traditional Use and Pharmacological Activity of *Picrasma quassioides*: A Critical Reviews

**DOI:** 10.3390/nu12092584

**Published:** 2020-08-26

**Authors:** Muhammad Daniel Hakim Mohd Jamil, Muhammad Taher, Deny Susanti, Md Atiar Rahman, Zainul Amiruddin Zakaria

**Affiliations:** 1Department of Pharmaceutical Technology, Kulliyyah of Pharmacy, International Islamic University Malaysia, Kuantan 25200, Malaysia; hakimdaniel656@gmail.com; 2Department of Chemistry, Kulliyyah of Science, International Islamic University Malaysia, Kuantan 25200, Malaysia; 3Department of Biochemistry and Molecular Biology, University of Chittagong, Chittagong 4331, Bangladesh; atiar@cu.ac.bd; 4Department of Biomedical Science, Faculty of Medicine and Health Sciences, Universiti Putra Malaysia, UPM Serdang 43400, Selangor, Malaysia

**Keywords:** *Picrasma quassioides*, *Simaroubaceae*, anti-inflammatory, antioxidant, anti-malarial, antifeedant, anti-cancer, anti-ulcer, quassinoids

## Abstract

*Picrasma quassioides* is a member of the Simaroubaceae family commonly grown in the regions of Asia, the Himalayas, and India and has been used as a traditional herbal medicine to treat various illnesses such as fever, gastric discomfort, and pediculosis. This study aims to critically review the presence of phytochemicals in *P. quassioides* and correlate their pharmacological activities with the significance of its use as traditional medicine. Data were collected by reviewing numerous scientific articles from several journal databases on the pharmacological activities of *P. quassioides* using certain keywords. As a result, approximately 94 phytochemicals extracted from *P. quassioides* were found to be associated with quassinoids, β-carbolines and canthinones. These molecules exhibited various pharmacological benefits such as anti-inflammatory, antioxidant, anti-cancer, anti-microbial, and anti-parasitic activities which help to treat different diseases. However, *P. quassioides* were also found to have several toxicity effects in high doses, although the evidence regarding these effects is limited in proving its safe use and efficacy as herbal medicine. Accordingly, while it can be concluded that *P. quassioides* may have many potential pharmacological benefits with more phytochemistry discoveries, further research is required to determine its real value in terms of quality, safety, and efficacy of use.

## 1. Introduction

*Picrasma quassioides* (D.Don) Bennett, is a small tree or shrub, commonly referred to as “nigaki” bitterwood, that belongs to the *Simaroubaceae* family that houses well-known genera such as *Simarouba*, *Quassia* and 30 other members [1].

Historically, *P. quassioides* plants have been exploited as traditional medicine, with the stem and wood bark of these plants in particular used to treat various kinds of illnesses such as fever, malaria [2,3,4], gastritis [5,6], pediculosis [7], etc. The crude extracts contain various beneficial phytochemicals, including alkaloids and polyphenols, which are responsible for a variety of pharmacological activities ranging from anti-inflammatory to anti-parasitic activities, as observed in recent scientific discoveries.

Accordingly, this review aims to analyse and correlate the phytochemicals contained in this plant along with their pharmacological effects in regard to its significance as traditional medicine. This review covers the details related to this plant such as taxonomy, botanical description, geographical distribution, known traditional uses, significant phytochemicals, their pharmacological activities, and possible toxicity.

## 2. Methods

The information related to *P. quassioides* was gathered and analysed from numerous research papers and reviewed articles accessed from various online scientific databases such as Google Scholar, ScienceDirect and Scopus, using particular keywords related to the title. The keywords used in the search included “*Picrasma quassioides”*, “*Simaroubaceaea*”, “anti-inflammatory”, “antioxidant”, “anti-malarial”, “antifeedant”, “anti-cancer”, and “anti-ulcer”.

The molecular structures of phytochemical compounds were drawn using ChemDraw Ultra 12.0 software. The molecular structure and formula systematic naming for each available molecule referred to the International Union of Pure and Applied Chemistry (IUPAC) standards for organic chemistry from the PubChem database. The phytochemicals were categorised based on their respective skeleton structure.

## 3. Botanical Description, Taxonomy and Geographical Distribution of *Picrasma quassioides*

### 3.1. Botanical Description and Taxonomy

*P. quassioides* (D.Don) Bennett is a deciduous bitterwood tree, where its leaves are shed every fall or autumn season annually, and it grows up a the maximum height of 10 metres [8]. Its leaves are pinnate and are arranged discretely to facilitate shedding, along with alternate petioles at each stem growing at odd numbers at the terminal. The leaves are described as broadly ovate, with the average length of the mature leaf typically ranging between 15 and 30 cm, while its leaflet is measured within 9–15 cm. The flowers are dioecious, where the male and female sex organs are found in separate individual plants, grown by auxiliary cymes appearing with 4–5 oblong or oval yellow-brown sepals in which the petals are grown in similar numbers as the sepals. The ripened fruits appear to be blue-green in colour measuring 6–8 mm in length. The tree bark appears reddish-brown, marked with yellow spots having a hard and rough texture.

*P. quassioides* is known by many names, especially within the Asian and Indian regions. The Japanese locals refer to this plant as “nigaki” [9], while in Chinese and Korean languages, it is referred to as “ku shu” [8] and “so-tae-na-moo” [10], respectively. Moreover, the Indians refer to this plant as “quassia” or “bharangi”. Figure 1 and Figure 2 show the images of leaves and wood bark of the plant [11].

*P. quassioides* is classified under the *Simaroubaceae* family, which houses approximately 30 other genera, mostly known to include the genera of *Quassia* and *Simarouba*. This is mostly due to the foundation of the special phytochemical group known as quassinoids, which are responsible for exhibiting a plethora of pharmacological activities ranging from anti-inflammatory to anti-parasitic activities [1]. At the next level down, it is classified into the genera of *Picrasma*, which is narrowed down further to between 6 to 10 known species. This plant is also synonymous to some of its family members such as *Picrasma ailanthoides* (Bunge) Planch., *Picrasma japonica* A. Gray, and *Simaba quassioides* D.Don [12].

### 3.2. Geographical Distribution

*P. quassioides* is rarely cultivated but instead grows in the wild. Its main habitat is in hill forest areas, grown among other hill trees such as fir, oak, deodar, etc. which reside at an altitude between 1800 and 2400 m above sea level [9]. However, some plants can also grow in mountainous mixed forest areas at a higher altitude, between 1400 to 3200 m above sea level. While this tree is well-distributed in East Asia (especially in Japan and Korea) and the Himalaya region, it also grows among the mountainous terrains of the central North and South-East region of China and neighbouring regions such as Taiwan, Tibet, India, Kashmir, Nepal, Sri Lanka and the Himalayas [8,12].

## 4. Phytochemicals in *Picrasma quassioides*

Numerous kinds of beneficial phytochemicals can be extracted from *P. quassioides* from various parts of the tree, such as its fruits, leaves, stem and wood bark, which can be categorised under certain groups, including quassinoids (**1**–**16**), β-carbolines (**17**–**24**), canthinones (**25**–**29**), neolignans (**30**–**32**), apotirucallanes (**33**–**36**), tirucallanes (**37**–**43**) and cinnamamides (**44**), which are based on each respective skeleton structure. The majority of the phytochemicals found from the literature belonged to quassinoids, due to the specificity of this tree’s family taxonomy. The examples of available phytochemicals along with their classifications and extracted sources are listed in Table 1.

## 5. Traditional Uses of *Picrasma quassioides*

### 5.1. Use for Treating Gastric Discomfort

Historically, *Picrasma quassioides* has been used to treat stomach discomfort, given its phytochemicals that produce a distinct bitter taste. It is an extremely well-known traditional medicine used among Koreans for treating chronic dyspepsia, which is uncontrollable gastric acid secretion, by helping to prevent ulceration [10]. Besides its gastro-protective effect on inhibiting gastric acid secretion, it also helps to facilitate mucosa production and speeds up tissue regeneration [5], while its bitter taste also helps to restore one’s appetite and improve digestion [9].

### 5.2. Use for Pediculosis

*Picrasma quassioides* is also used for treating head lice infestation that commonly targets children and women. The stem bark of *Picrasma quassioides* is ground into a powder before applying topically on the affected area [7]. This quassia powder does not only inhibit the growth of the parasite but also helps to reduce the irritation. It has also been proven to involve the use of the quassia tincture in pediculosis treatment among 454 patients, producing effective results with 99% of the patients treated successfully [31].

### 5.3. Use in Treatment of Fever and Infection

Since ancient times, *Picrasma quassioides* has been used as one of the important ingredients in traditional Chinese medicine, *kumu* injection therapy, as recorded in Chinese pharmacopoeias. This therapy manages to alleviate the symptoms of fever through detoxification and heat-clearing mechanism [32]. Besides *kumu* therapy, it also possesses antibacterial benefits, which helps to treat infections such as the common cold, upper respiratory tract infection and diarrhoea. This has been proven by several research studies conducted in China to assess the efficacy of this medicine in treating various infections among children [32].

## 6. Pharmacological Activities of *Picrasma quassioides*

*Picrasma quassioides* displays various potential pharmacological activities thanks to its phytochemicals, such as anti-inflammatory, antioxidants, anti-cancer, anti-microbial and anti-parasitic mechanisms. The effectiveness is contributed through the medicinal chemistry and mechanism of the actions of compounds towards their respective targets.

### 6.1. Anti-Inflammatory Properties

The anti-inflammatory system encompasses the intervention of the inflammatory pathway, thus preventing further inflammatory damage and reversing the pathological conditions. The general anti-inflammatory mechanism of *P. quassioides* can be described through several pathways. First, the anti-inflammatory mechanism of *P. quassioides* can be explained through interrupting inflammatory gene regulation pathways such as nuclear factor NF-κB and extracellular signal-regulated kinase (ERK) phosphorylation in LPS-induced macrophages, thereby suppressing the cyclooxygenase 2, cyclooxygenase 2 (COX-2) activity [33].

Second, its anti-inflammatory activity also involves suppressing the expression of pro-inflammatory cytokines, as proven through an in vitro study involving the quassidines, which managed to restrict the production of pro-inflammatory cytokines such as NO, TNF-α and IL-6 by lipopolysaccharides (LPS) in a mouse monocyte–macrophage RAW 264.7 with the inhibitory concentration of IC_50_ of 89.39–100.00, 88.41 and >100 μM, respectively, resulting in a good efficacy level when compared to the conventional anti-inflammatory corticosteroid, hydrocortisone [34].

Furthermore, *P. quassioides* also manages to limit the sufficiency of necessary supplies for the activity of inflammatory cells, such as macrophage and neutrophils, thereby minimising inflammation. For example, 1-hydroxymethyl-8-hydroxy-β-carboline (**17**) in *P. quassioides* was found to inhibit angiogenesis with the lowest observed effective concentration (LOEC) of 5 μM when tested in zebrafish with minimal toxicity [17], which helps to limit sufficient oxygen and nutrient supply for the inflammatory tissues, thereby reducing inflammation.

#### 6.1.1. Gastric Dyspepsia Treatment

Gastritis refers to the inflammatory damage of the gastric mucosa lining due to uncontrollable gastric acid secretion by gastric parietal cells caused by several intrinsic and extrinsic factors, such as hypersecretion syndrome, drugs, especially non-steroidal anti-inflammatory (NSAIDs), *Helicobacter pylori* infection, etc. [35] This condition leads to persistent inflammatory mucosa injury which interferes with the gastric mucosa protective function and tissue regeneration.

The gastric acid-suppressive mechanism of *P. quassioides* can be explained through the canthinone compounds such as nigakinone (**25**) and methylnigakinone (**26**) which inhibit the release of histamine into gastric parietal cells, thereby limiting gastric acid secretion and mucosa damage via acidic corrosion [36]. Besides, these canthinones also stimulate more prostaglandin secretion for increasing the production of alkaline mucus from goblet cells for neutralisation and protection against the acidity and releasing more growth factors for tissue healing processes [5,6,37]. Hence, these mechanisms can help to reduce the incidence of ulcerative gastritis.

The efficacy of *P. quassioides* on gastric ulceration has also been evaluated through its tissue-healing properties, using the aqueous extract of *P. quassioides* which successfully reduced gastric lesion sizes by 47.71% and 72.24% and suppressed gastric acid secretion at 29.92% and 63.63%. This was measured by the change in pH value when administered at a 200 and 400 mg/kg dose, respectively, suggesting that the efficacy acted in a dose-dependent manner [38].

#### 6.1.2. Anti-Hypertensive Effect

One of the possible causes of hypertension is atherosclerosis, which refers to the hardening of the arteries due to clogging or accumulation of plaque that continues to protrude in the lumen, increasing peripheral resistance and obstructing oxygenated blood flow to the tissues [39]. This condition can be extremely dangerous since it can cause insufficient oxygen supply to tissues, thereby limiting normal tissue metabolisms and functions, especially of the heart, essential organs, and other bodily systems. Notably, *P. quassioides* is capable of displaying anti-hypertensive properties through the prevention of atherosclerosis.

The components in *P. quassioides*, such as canthinones, β-carbolines and Picrasmalignan A (**30**), are responsible for suppressing LPS-induced macrophage activity by inhibiting the induced nitrous oxide synthase (iNOS) activity, resulting in limiting the NO over-secretion [25]. High NO concentration can be dangerous despite the fact that its vasodilation property as an abnormality can trigger the inflammation process. Through this inhibition, the release of pro-inflammatory cytokines, such as TNF-α, IFN-γ, IL-6, etc., can be prevented, therefore minimising the inflammatory damage on endothelial cells, especially by macrophages [25,33,40]. The anti-inflammation activity also manages to limit macrophage activity deposited in blood vessels and platelet aggregation on the injured site, thus minimising atherosclerotic plaque formation and occlusion in the lumen.

*P. quassioides* also possesses some vasoprotective properties, which manage to regulate the levels of NO by increasing the level of the natural antioxidant, superoxide dismutase enzyme (SOD), in a dose-dependent pattern. This subsequently helps to prevent NO degradation by free reactive oxygen species (ROS) radicals with the increase in the level of NO in the same manner [41]. In other words, it helps with controlling vasodilation to improve blood flow. The stem extract of *P. quassioides* also possesses anti-thrombotic activity, which delays clotting time and reduces the thrombotic mass in clogged arteries through suppressing secondary coagulation, as observed when tested in FeCl_3_-induced thrombotic rats [42].

#### 6.1.3. Asthma Treatment

Asthma refers to an obstructive airway disease that involves the swelling of the airway tract, especially bronchioles due to inflammation, triggered by an allergen (hypersensitivity). This condition causes narrowing of the airway tract which can cause difficulty in breathing; however, it is reversible with proper treatment [43].

Canthinone and β-carboline molecules contained in *P. quassioides* help to relieve asthma through suppressing the release of pro-inflammatory cytokines. This was proven through an in vivo study carried out on allergy-induced asthmatic mice, where the mice were administered with 15 and 30 mg/kg of the aqueous extract. The results obtained found that the inflammatory cytokine count, such as interleukins IL-4, IL-5, IL-13 and immunoglobulin E, IgE, significantly decreased in a dose-dependent manner inversely proportional with improved histological observations, i.e., less tissue remodelling compared to the normal control group [44].

Furthermore, the study also found that the inhibition of inflammatory cytokines of *P. quassioides* also leads to the declination of other mediators such as iNOS, which limits the over-secretion of NO, subsequently suppressing the migration of inflammatory cells to the bronchial and/or alveolar epithelium lining, reducing swelling of the epithelial and excessive mucus production that can obstruct the airway tract [22,44,45]. As such, this helps to overcome airway hyper-responsiveness, resulting in improving the respiratory airway tract for better breathing.

### 6.2. Antioxidant Activity

Oxidative stress refers to the disturbed balance between the concentration of free-radical molecules and antioxidants in the tissues, which can pose harm towards normal health tissues through oxidative cellular damage by ROS [46]. Antioxidants are specific molecules that inhibit the production or depleting free-radical molecules, especially ROS, therefore helping to prevent cellular damage, ensuring cell survivability [47]. Notably, antioxidant activity is closely related to anti-inflammation.

The antioxidant property of *P. quassioides* has also been investigated using its methanol and ethanol extracts at a concentration of 100 μg/mL, which managed to suppress superoxide radicals by 50.50% and 47.90%, respectively, while suppressing hydroxyl radicals by 20% and 30%, respectively [48].

#### 6.2.1. Osteoporosis Prevention

Osteoporosis refers to a bone condition where the density of the mineralised bone matrix is greatly reduced, causing declination in mechanical strength to support body weight and increasing bone fracture, leading to severe pain during movement and difficulty to stand, as well as increasing the risk of fracture upon impact [49]. One of the possible causes behind osteoporosis is the activity of the uncontrolled osteoclasts.

*P. quassioides* is able to exhibit a different inhibitory effect on osteoporosis in two concentrations, where at a low concentration of 0.1 mg/mL, it only caused staining of the osteoclasts in the bone matrix culture. In contrast, at a higher concentration of 0.2 mg/mL, it managed to suppress osteoclast proliferation and activity [50].

Another study also explained the actual mechanism behind the bone protective property of *P. quassioides* through the Picrasidine I (**21**) compound, which scavenged free radicals such as peroxide radicals and H_2_O_2_, thus preventing its release to osteoblasts and thereby preventing the activation and binding of the receptor activator of nuclear factor κB ligand (RANKL)to the cell surface receptor RANK. Accordingly, this resulted in the inhibition of two different downstream pathways in osteoclastogenesis, which are the nuclear factor NF-κB pathway and mitogen-activated protein kinases (MAPKs) in the osteoclast precursor cells, consequently leading to the suppression of two main osteoclastogenesis transcription factors, c-FOS and NFATc1 [51]. Thus, the suppression of these pathways prevents the osteoclastogenesis process, thereby limiting the differentiation and replication of mature osteoclasts. Picrasidine I (**21**) also limits ROS release from mature osteoclasts, thereby limiting the digestion of bone matrix, preventing the bone resorption process and reducing the risk of osteoporosis [51].

#### 6.2.2. Neuroprotective Activity

*P. quassioides* exhibits a neuroprotective function that prevents oxidative stress damage of nerve tissues. For example, Alzheimer’s disease is one of the neurodegenerative diseases which relates to the degeneration of brain cells and heavily impairing the memory and coordinating capability, which results in the declination of an individual’s thinking, behaviour and conversation capability [52].

The neuroprotective property of *P. quassioides* derives from its phytochemical, Picrasidine O (**27**), in preventing further brain damage due to glutamate toxicity. This is achieved by reducing the amount of accumulating glutamate in the brain, hence, limiting the hyper-excitatory stimulation of the neural tissue by the neurotransmitter glutamic acid and kainic acid through antagonistic inhibition, thereby potentially reducing the risk of cerebral ischaemia [24]. Glutamate toxicity occurs when glutamate binds to the N-methyl-d-aspartate (NMDA) receptors in the brain cells to increase neuro-excitability for faster impulse transmission. However, the excessive concentration or toxicity of glutamate can cause an increased influx of calcium ion, leading to increased free-radical production and brain cell damage [53].

Another phytochemical that provides neuroprotective ability is kumulactone A (**2**), which protects normal healthy nerve cells by preventing free radical-induced cellular apoptosis by scavenging H_2_O_2_ radicals, and subsequently limiting the caspase-3 activity by approximately 25% at a dose of 25 μM against 300 μM of H_2_O_2_, thereby helping to prevent nerve cell apoptosis [13].

*P. quassioides* also provides additional neuroprotective properties through dehydrocrenatidine (**18**), which exhibits analgesic property by limiting neuronal excitability in a concentration-dependent pattern via the inhibition of the tetradotoxin-resistant (TTX-R) and sensitive ( TTX-S) voltage-gated sodium channel when tested in rat ganglionic neurons, therefore limiting the enhanced sodium ion influx into the nerve cells slowing down the action potential and impulse transmission in neuropathic pain [18]. Consequently, the limitation of this excitability helps to reduce pain sensation, especially in dementia-type patients, e.g., those with Alzheimer’s.

Notably, the enhanced potency of the neuroprotective property in β-carboline molecules can be related to the structural activity relationship, that is, the addition of alkoxy groups to alkyl chains, especially in 6,12-dimethoxy-3-ethyl-β-carboline (**22**), which facilitates the suppression of acetylcholinesterase (AChE) activity, therefore preventing the risk of Alzheimer’s disease [20].

### 6.3. Anti-Cancer Activity

Cancer normally refers to abnormal cell growth outside its regular cell cycle, resulting in uncontrollable, abnormal cell replication that spreads to other adjacent normal cells due to the mutation in the specific DNA sequence of the cells that are responsible for the regulation of normal cell growth and apoptosis, mainly proto-oncogenes, tumour suppressor genes and DNA repair genes [54]. Usually, defective or damaged cells are programmed to be terminated via apoptosis; however, these abnormal cells are able to resist apoptosis, keep growing and compete with normal cells for sufficient supply.

The anti-cancer property of *P. quassioides* has been tested in various in vitro studies using different cancer cell samples. For example, the aqueous extracts of *P. quassioides*, such as methanol (MeOH) and ethanol (EtOH) extracts, are very potent as anti-cancer agents that work via cancer cell necrosis and apoptosis, producing a cytotoxicity level of between 50 to 60% against NCI-N87 cancer cells when administered at a dose of 1 mg/mL each. However, only the apoptosis mechanism is considered in the anti-cancer evaluation [48].

On the other hand, picraquassin B (**34**) and kumuquassin C (**43**) are the most potent apotirucallane and tirucallane compounds, respectively, that display anti-cancer activity through stimulating the mitochondrial apoptotic pathway. This promotes cytochrome c release for caspase-3 activation and increases pro-apoptotic protein expression, which leads to cancer cell apoptosis when tested against MKN-28 and A-549 cells with IC_50_ of 2.5 and 5.6 μM and HepG2 cancer cells with IC_50_ of 21.72 μM, respectively [28,29].

Similarly, quassinoid, β-carboline, and canthinone molecules in *P. quassioides* also exhibit potent anti-cancer activity in various cancer cell lines in animal trials using the same mentioned mechanism. In an in vitro test, Bruceantin (**1**) was used against RPMI-8226 cancer cells producing an effective IC_50_ of 2.5 and 5.0 mg/kg against early and advance tumours, respectively. Moreover, dehydrocrenatidine (**31**) was effective against A2780 cells with an IC_50_ of 2.02 ± 0.95 μM and against SKOV3 cells at an IC_50_ of 11.89 ± 2.38 μM. Meanwhile, nigakinone (**25**) and methylnigakinone (**26**) were also effective against HepG2 cancer cells [3,55,56].

The stereoisomerity of phytochemicals also influences the anti-cancer property of *P. quassioides* (refer to Figure 3), especially involving the enantiomers of neolignan molecules such as dehydrodiconiferyl alcohol (**31**), in which the enantiomer (+)-7*S*, 8*R* has better enantio-selectivity in regard to the cytotoxic activity towards HepG2 cancer cells than enantiomer (+)-7*R*, 8*S*. This was proven by the IC_50_ values of 35.6 μM and 104.4 μM, respectively, indicating that the configuration also affects the cytotoxic performance against cancer cells via apoptosis and ROS generation [26].

Furthermore, quassidines I (**19**) and J (**20**) extracted from *P. quassioides* exist as a racemic mixture, while the cytotoxicity of (+)-S enantiomers for both quassidines was found to be more potent cytotoxic compared to (-)-R enantiomers when tested against human cervical HeLa cancer cells and MKN-28 cell lines with an IC_50_ of 5.75 and 6.30 μM, respectively, for quassidine I (**19**), and 4.03 and 4.91 μM, respectively, for quassidine J (**20**) [19]. Kumudine B (**24**) also possess enantiomers, among which the (-)- enantiomer is superior to (+)- enantiomer regarding apoptotic induction of Hep3B cancer cells, producing 60.0% and 20.1% potency with an IC_50_ of 25 μM [21].

### 6.4. Anti-Microbial Effect

*P. quassioides* is also able to exhibit anti-microbial activities towards both bacterial and fungal species. Here, the essential oil of *P. quassioides* has intermediate anti-microbial action in inhibiting the growth of bacterial and fungal pathogens proven through a study which tested against the test microbe culture *Bacillus subtilis* bacterium and *Ganoderma lucidum* fungus, which produced the inhibition activity of a 12 mm^2^ inhibition area for both antibacterial and antifungal activity, respectively [57].

Notwithstanding, another anti-microbial property of *P. quassioides* was also tested using the β-carboline molecules against test microbes *Staphylococcus aureus*, *Escherichia coli*, *Candida albicans* and *Aspergillus niger*, producing a minimum inhibition concentration (MIC) of 32–64, 4–8, 16–64, and 16–32 μg/mL, respectively, which indicates moderate anti-microbial activity [20]. The presence of the alkyl chain at the C-3 position of the β-carboline ring, especially 6,12-dimethoxy-3-ethyl-β-carboline (**22**), enhanced the anti-microbial efficacy [20], which suggests that *P. quassioides* might be potentially used as an anti-infective agent.

### 6.5. Anti-Parasitic Capability

The anti-parasitic property of *P. quassioides* has also been applied as a treatment agent in two different situations, namely, malaria and pediculosis.

#### 6.5.1. Anti-Malarial Property

Malaria is one of the deadly epidemic diseases caused by *Plasmodium* parasites, especially *Plasmodium falciparum*, which are transmitted through mosquito bites of infected female *Anopheles* mosquitoes that act as malarial vectors [58,59]. Even though this disease is treatable and preventable, if left untreated, it can lead to severe complications.

The phytochemical simalikalactone D (**8**), which was extracted from the young and mature tea leaves of *Quassia amara*, also found in most *Picrasma species*, acts as the most potent anti-malarial compound. This was proven from an in vitro test that produced complete inhibition activity of the strains of *Plasmodium falciparum* at the inhibition concentration with an IC_50_ of 10 nM, while the other extracted molecules produced very insignificant activity [2]. Another study had proven the conclusion of a previous study, mentioning that the anti-plasmodial mechanism of simalikalactone D (**8**) was achieved by inhibiting *Plasmodium* protein synthesis, especially during the DNA replication stage in mature *Plasmodium* cells at a concentration of 2 μg/mL in an in vitro test [3]. However, simalikalactone D (**8**) was only shown to be effective in the in vitro test, while simalikalactone E (**9**) was effective in the in vivo test [3].

Another anti-malarial study assessing β-carboline activity of *Plasmodium* parasite inhibition showed that the modification of the available neutral β-carboline into N-alkyl-β-carbolinium salts invariably leads to enhanced anti-malarial activity (refer to Figure 4) by producing a lower effective concentration of EC_50_ ranging between 1.3 × 10^-7^ and 1.3 × 10^-5^ M, while the previous EC_50_ was within 5.0 × 10^-6^ and 3.1 × 10^-5^ M [4]. Further, this modification also reduces toxicity risk of normal healthy cells by increasing its selectivity, ranging from 0.76–1.0% to 48–65%. However, its selective toxicity is still lower compared to the conventional anti-malarial drug, quinine, although both have comparable anti-malarial efficacy. This can be contributed through the presence of a π-delocalised lipophilic cationic structure in modified β-carbolinium salts which enhances their potency and selective toxicity [4].

#### 6.5.2. Pediculosis Treatment

As mentioned previously, *P. quassioides* has been used to treat head lice or pediculosis caused by ectoparasite *Pediculus humanus capitis*. It is prepared from the tree stem, which is separated from its bark and chipped into small pieces. The wood chips are then dusted into a powder form and applied topically on the scalp to prevent lice infestation, thus, overcoming the infestation and reducing irritation [7]. In modern times, quassia tinctures or liquid extracts have been used as an individual product or formulated in hair shampoo formulations that are used instead of powders for topical application for head lice treatment. However, at present, there are insufficient scientific records related to the assessment of this traditional therapy, but the efficacy remains acceptable in current day. The possible explanation behind the efficacy can be proven through the property exhibited by the phytochemicals of this tree, such as the quassins [7].

In later research studies, quassin efficacy behind pediculosis treatment was further investigated through the antifeedant mechanism against pest insects. The research described the release of secondary metabolite chemicals that affect the central nervous system, such as the feeder, which renders disabled movement but does not kill the pests [60,61]. However, this study was not carried out in clinical trials but instead through the assessment of bio-pesticides to which the same mechanism applies. Therefore, *P. quassioides* wood chip powder is perceived as prophylactic rather than as an insecticide agent, which also coincides with the previous statement [62].

## 7. Possible Toxicity

While *P. quassioides* has successfully displayed its pharmacological effects in various in vivo and in vitro studies safely, there are some studies, although limited, that prove it can pose some toxicity and adverse effects, particularly during in vivo studies.

An example of a toxicity study is a clinical study on the maximum optimum dosage for simalikalactone D (**8**) in the form of quassia herbal tea in anti-malarial therapy, which exceeds the highest daily dose of 75 μg/kg, resulting in various side effects such as hypotension and nausea. However, doubling the maximum amount resulted in thrombocytopenia [2]. In contrast to human clinical trials in anti-cancer assessment, bruceantin (**1**) neglected to show positive results, unlike in the previous animal trials. Instead, it produced similar adverse effects to simalikalactone D (**8**) at a higher dose in humans [3].

Further evidence of toxicity has also been reported in an anti-cancer study involving the methanol and ethanol extracts of *P. quassioides* having some cytotoxicity effect on normal kidney cells, possibly causing acute kidney injury [48]. However, the study neglected to mention the actual maximum tolerated amount of the extracts to avoid this effect.

In another study, the nigakinone (**25**) of *P. quassioides* was reported to induce toxicity during an in vivo test involving a zebrafish embryo, where the exposure led to increased copper chelation on the copper transport proteins, thereby causing elevated gene expression related to oxidative stress. This could pose further damage towards the zebrafish embryo, especially when administered at a dose of 7.5 μM [63]. However, this cytotoxic effect could not be applied to human tissues, given the absence of human tissue culture as the sample for this cytotoxicity study.

## 8. Limitation of the Studies

From the analysed research articles on the pharmacological benefits of this plant, it was discovered that there was limited evidence on its true efficacy and toxicity towards the human body. Some toxicity studies on this herb were found to be insufficient to determine the actual safe value as herbal medicine since the occurrence of the adverse advents were relatively unpredictable.

Furthermore, the majority of the pharmacological effects exhibited by its phytochemicals were only observed and evaluated through in vitro and in vivo methods on animal models where the benefits were extrapolated to human physiology, rather than being tested on actual human subjects in clinical trials. Moreover, there is limited insight regarding the adverse reactions, possible drug–drug or drug–food interactions with this herb and other considerations such as safety regarding its usage during pregnancy.

Therefore, the suggestion in solving this issue is to conduct more detailed pharmacological studies which extend to the pharmacokinetic and pharmacodynamics aspects to gain further understanding of this plant’s true value in terms of efficacy, quality, and safety.

## 9. Conclusions

*P. quassioides* provides potential pharmacological benefits to consumers due to the discovery of its phytochemicals. The significance of its traditional uses has been well-established through scientific studies. However, while there are numerous limitations and weaknesses in these research studies in proving its quality, efficacy, and safety, the plant continues to be used. Despite this, its usage cannot completely replace the use of modern medicines in disease treatment. Extensive discovery of the structure–activity relationships of these phytochemicals would indeed help to enhance the medicinal potency while minimising toxicity when considering the multiple behaviours of these molecules (such as stereoisomerism, structural complexity, etc.), which can be challenging in synthesising or modifying them into ideal alkaloid phytochemicals.

## Figures and Tables

**Figure 1 nutrients-12-02584-f001:**
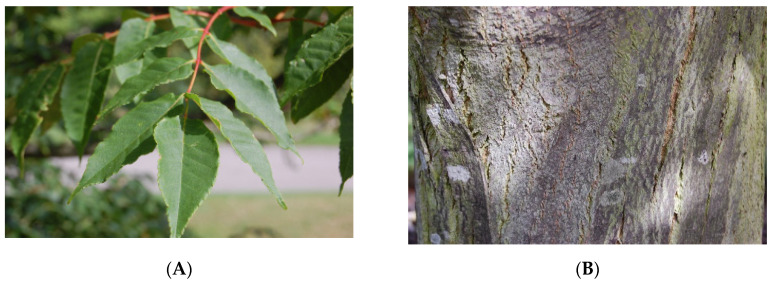
Close-up image of *Picrasma quassioides* (D.Don) Bennett tree: (**A**) leaves (**B**)wood bark. The images were obtained from Reference [11] with permission.

**Figure 2 nutrients-12-02584-f002:**
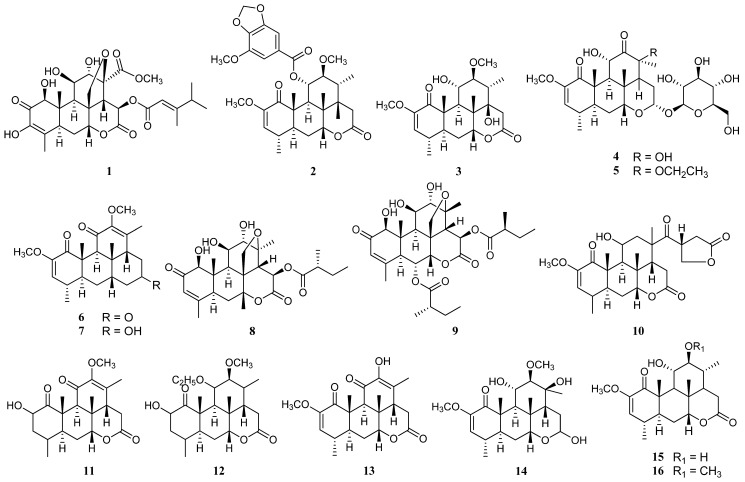
The molecule structures of the phytochemicals available in *Picrasma quassioides*.

**Figure 3 nutrients-12-02584-f003:**
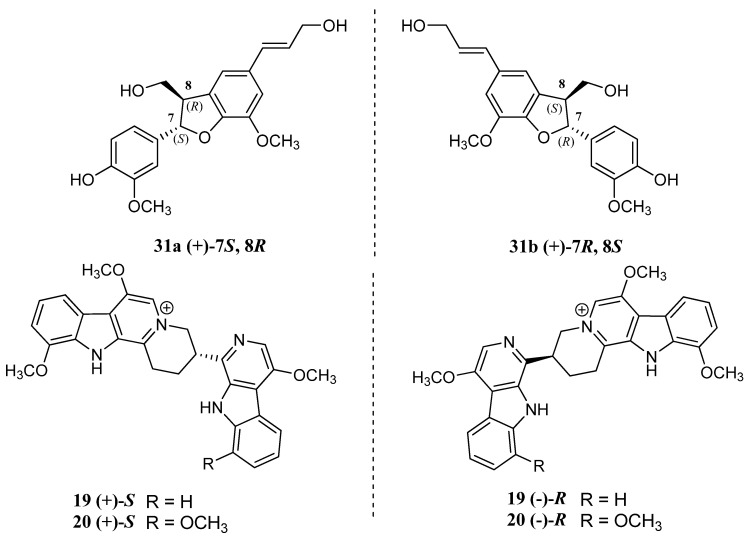
The stereoisomerity of dehydrodiconiferyl alcohol (**31**), quassidine I (**19**) and J (**20**) and kumudine B (**24**) (from left to right) responsible for the enhanced anti-cancer property of *Picrasma quassioides* [19,21,26].

**Figure 4 nutrients-12-02584-f004:**
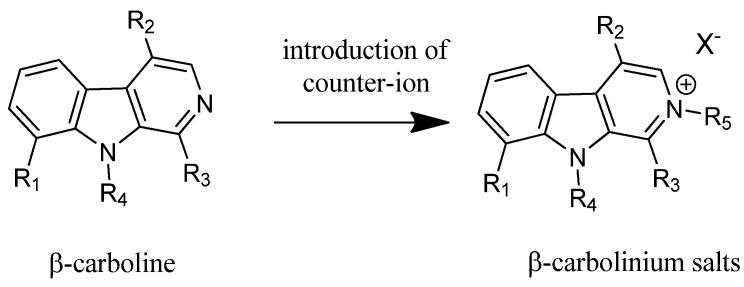
The modification of general β-carboline compounds into the β-carbolinium salts for enhanced anti-malarial activity of *Picrasma quassioides* [4].

**Table 1 nutrients-12-02584-t001:** Phytochemicals available in *Picrasma quassioides*.

Compound No.	Molecular Group	Phytochemical Name	Molecular Formula	Extract Source	References
**1**	Quassinoids	Bruceantin	C_28_H_36_O_11_	Stem	[3]
**2**		Kumulactone A	C_31_H_38_O_10_	Stem	[13]
**3**		Kumulactone B	C_22_H_32_O_7_	Stem	
**4**		Picrasinoside J	C_27_H_40_O_12_	Stem	
**5**		Picrasinoside K	C_29_H_42_O_13_	Stem	
**6**		Quassine	C_22_H_28_O_6_	Wood bark	[7]
**7**		Neoquassine	C_22_H_30_O_6_	Wood bark	
**8**		Simalikalactone D	C_25_H_34_O_9_	Tea leaves	[2,3]
**9**		Simalikalactone E	C_30_H_42_O_11_	Tea leaves	
**10**		Picrasin A	C_26_H_34_O_8_	Stem	[14]
**11**		Picrasin B	C_21_H_28_O_6_	Stem	
**12**		Picrasin C	C_23_H_34_O_7_	Stem	
**13**		12-hydroxyquassin	C_22_H_28_O_7_	Leaf, twigs	[15,16]
**14**		Nigakihemiacetal A	C_22_H_34_O_7_	Leaf, twigs	
**15**		Nigakilactone A	C_21_H_30_O_6_	Leaf, twigs	
**16**		Nigakilactone B	C_22_H_32_O_6_	Leaf, twigs	
**17**	β-carbolines	1-hydroxymethyl-8-hydroxy-β-carboline	C_12_H_10_N_2_O_2_	Stem	[17]
**18**		Dehydrocrenatidine	C_15_H_14_N_2_O_2_	Stem	[18]
**19**		Quassidine I	C_29_H_27_N_4_O_3_	Stem	[19]
**20**		Quassidine J	C_30_H_29_N_4_O_3_	Stem	
**21**		Picrasidine I	C_14_H_14_N_2_O_2_	Stem	
**22**		6,12-dimethoxy-3-ethyl- β-carboline	C_15_H_16_N_2_O_2_	Stem	[20]
**23**		Kumudine A	C_23_H_22_N_2_O_4_	Stem	[21]
**24**		Kumudine B	C_28_H_22_N_4_O_5_	Stem	
**25**	Canthinones	5-hydroxy-4-methoxycanthin-6-one (Nigakinone)	C_15_H_10_N_2_O_3_	Bark	[6,22,23]
**26**		4,5-dimethoxycanthin-6-one (Methylnigakinone)	C_16_H_12_N_2_O_3_	Bark	[6,23]
**27**		Picrasidine O	C_16_H_12_N_2_O_3_	Stem	[24]
**28**		4,5-dimethoxy-10-hydroxycanthin-6-one	C_16_H_12_N_2_O_4_	Wood bark	[23]
**29**		8-hydroxycanthin-6-one	C_14_H_8_N_2_O_4_	Wood bark	
**30**	Neolignans	Picrasmalignan A	C_30_H_30_O_9_	Stem	[25]
**31**		Dehydrodiconiferyl alcohol	C_20_H_22_O_6_	Stem	[26]
**32**		Picraquassioside C	C_28_H_38_O_14_	Fruits	[27]
**33**	Apotirucallanes	Picraquassin A	C_30_H_50_O_5_	Stem	[28]
**34**		Picraquassin B	C_32_H_52_O_4_	Stem	
**35**		Picraquassin C	C_37_H_59_O_9_	Stem	
**36**		Picraquassin D	C_30_H_50_O_5_	Stem	
**37**	Tirucallanes	Picraquassin E	C_31_H_50_O_3_	Stem	
**38**		Picraquassin I	C_31_H_53_O_5_	Stem	
**39**		Picraquasssin J	C_31_H_53_O_5_	Stem	
**40**		Picraquassin K	C_32_H_54_O_5_	Stem	
**41**		Kumuquassin A	C_30_H_44_O_5_	Stem	[29]
**42**		Kumuquassin B	C_30_H_46_O_6_	Stem	
**43**		Kumuquassin C	C_30_H_45_O_5_	Stem	
**44**	Cinnamamides	Picrasamide A	C_21_H_21_N_2_O_6_	Stem	[30]

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
