# Peer review of "Phytochemistry, Traditional Use and Pharmacological Activity of Picrasma quassioides: A Critical Reviews"

_nutrients, 2020, doi:10.3390/nu12092584_

Round 1
Reviewer 1 Report
Article: „Review of Phytochemistry, Traditional Use and Pharmacological Activity of Picrasma quassioides”. The aim of the study was aims to analyse and correlate the phytochemicals contained in this plant along with their pharmacological effects towards its significance as traditional medicine. This review covers the details related to this plant such as taxonomy, botanical description, geographical distribution, known traditional uses, significant phytochemicals, their pharmacological activities, and possible toxicity. The article is really intresting, but needs some minor tweaks.
#line 22: „P. quassioides” should be intalic in all text.
#line 26: what does mean „aids”?
#line 100-107: align the text
#line 156: should be „…the cyclooxygenase 2 (COX-2) activity…”
Author Response
Reviewer 1
Comments and Suggestions for Authors
Article: „Review of Phytochemistry, Traditional Use and Pharmacological Activity of Picrasma quassioides”. The aim of the study was aims to analyse and correlate the phytochemicals contained in this plant along with their pharmacological effects towards its significance as traditional medicine. This review covers the details related to this plant such as taxonomy, botanical description, geographical distribution, known traditional uses, significant phytochemicals, their pharmacological activities, and possible toxicity. The article is really interesting, but needs some minor tweaks.
Response: Thank you very much for the positive comments.
#line 22: „P. quassioides” should be intalic in all text.
Response: All P.quassioides have been italicised
#line 26: what does mean „aids”?
Response: The sentence has been revised
#line 100-107: align the text
Response: Done
#line 156: should be „…the cyclooxygenase 2 (COX-2) activity…”
Response: Done

Reviewer 2 Report
The manuscript entitled “Review of Phytochemistry, Traditional Use and Pharmacological Activity of Picrasma quassioides” deals with the chemical characterization of Picrasma quassioides, a traditional herbal medicine used in folk and traditional medicine with the aim to treat different kinds of diseases. In addition to the phytochemical composition of this plant, the authors analyzed also its main pharmacological activities, linked to the presence of different classes of bioactive compounds.
The manuscript is generally clear and well written, apart from some minor errors (i.e. the authors do not always quote the Latin name of the plant in italics). It describes a thorough and well-described report, which summarizes the potential of Picrasma quassioides not only in modern medicine, but also in the dietary-supplement field.
The review is quite interesting, however some revisions are necessary.
- Title: I would remove "Review of" at the beginning of the title. If the authors want to specify the typology of the manuscript directly in the title, they could modify it as follows: “Phytochemistry, Traditional Use and Pharmacological Activity of Picrasma quassioides: a critical review” or “Picrasma quassioides: a review on its Phytochemistry, Traditional Use and Pharmacological Activity” or otherwise just leave “Phytochemistry, Traditional Use and Pharmacological Activity of Picrasma quassioides”
- Abstract: in addition to minor error (i.e. Simaroubaceae -> Simaroubaceae; Picrasma quassioides -> Picrasma quassioides, etc..) which are present throughout the manuscript, I think that different parts of the abstract are unnecessary. For example, at line 21 – 23 the authors simply described what a review consists of. I believe this part is unnecessary, and should be removed. For the same reason, remove at Line 23 “As a result”.
- Keywords: also here Simaroubaceae should be written in italics. The authors could add other keywords concerning the main families of chemical classes reported and discussed in this work.
- Introduction: this introduction is too short, and above all it does not address important issues. Some information regarding the economic importance and impact of this plant, should also be added in this section.
- Methods: this paragraph and the related information are not necessary. Please, remove it.
- Figure 1 and Figure 2 are not cited in the text.
- Line 107: it is not enough to write “table below”. Please insert the number of the table.
- Figure 1: please remove the brackets () from the panel title, and place the CAPITAL letter in the upper right corner of corresponding panel. Modify the caption as follows: “Figure 1: The close-up image of Picrasma quassioides (D. Don) Bennett tree: (A) leaves and fruits; (B) wood bark [11]”. Moreover, I want specify that the authors need the permission to publish photographs for which they do not hold copyright. Please, see https://www.mdpi.com/authors/rights for further information.
- Section 3: now section 2. It is better if the authors consider to rename the section as “Botanical Description, Taxonomy and Geographical distribution of Picrasma quassioides”.
- Subsection 2.3: Please, merge this section with that concerning the botanical description, and rename the title as following “Botanical Description and Taxonomy”.
- New subsection: I would insert an additional paragraph on the commercial importance of this plant.
- I think that figures 4-6 should be included in the manuscript, and immediately after being mentioned in the text.
- Section 3: rename as “Phytochemicals in Picrasma quassioides plants”
- Figure 2: according to me it is too big. I would advice the authors to divide the figure into other smaller figures which can be included separately on one-single page. For example, the authors could divide the figures according to family classes. (Fig. 2: Chemical structures of Quassinoid compounds identified in Stem, Wood bark and Leaves of Picrasma quassioides plants; Fig. 3: Chemical structures of β-carboline and Canthinone compounds identified by Stem, Wood and Bark of Picrasma quassioides plants; etc…). The authors should modify the relative captions.
The manuscript entitled “Review of Phytochemistry, Traditional Use and Pharmacological Activity of Picrasma quassioides” deals with the chemical characterization of Picrasma quassioides, a traditional herbal medicine used in folk and traditional medicine with the aim to treat different kinds of diseases. In addition to the phytochemical composition of this plant, the authors analyzed also its main pharmacological activities, linked to the presence of different classes of bioactive compounds.
The manuscript is generally clear and well written, apart from some minor errors (i.e. the authors do not always quote the Latin name of the plant in italics). It describes a thorough and well-described report, which summarizes the potential of Picrasma quassioides not only in modern medicine, but also in the dietary-supplement field.
The review is quite interesting, however a minor revision is necessary before the publication in Nutrients.
- Title: I would remove "Review of" at the beginning of the title. If the authors want to specify the typology of the manuscript directly in the title, they could modify it as follows: “Phytochemistry, Traditional Use and Pharmacological Activity of Picrasma quassioides: a critical review” or “Picrasma quassioides: a review on its Phytochemistry, Traditional Use and Pharmacological Activity” or otherwise just leave “Phytochemistry, Traditional Use and Pharmacological Activity of Picrasma quassioides”
- Abstract: in addition to minor error (i.e. Simaroubaceae -> Simaroubaceae; Picrasma quassioides -> Picrasma quassioides, etc..) which are present throughout the manuscript, I think that different parts of the abstract are unnecessary. For example, at line 21 – 23 the authors simply described what a review consists of. I believe this part is unnecessary, and should be removed. For the same reason, remove at Line 23 “As a result”.
- Keywords: also here Simaroubaceae should be written in italics. The authors could add other keywords concerning the main families of chemical classes reported and discussed in this work.
- Introduction: this introduction is too short, and above all it does not address important issues. Some information regarding the economic importance and impact of this plant, should also be added in this section.
- Methods: this paragraph and the related information are not necessary. Please, remove it.
- Figure 1 and Figure 2 are not cited in the text.
- Line 107: it is not enough to write “table below”. Please insert the number of the table.
- Figure 1: please remove the brackets () from the panel title, and place the CAPITAL letter in the upper right corner of corresponding panel. Modify the caption as follows: “Figure 1: The close-up image of Picrasma quassioides (D. Don) Bennett tree: (A) leaves and fruits; (B) wood bark [11]”. Moreover, I want specify that the authors need the permission to publish photographs for which they do not hold copyright. Please, see https://www.mdpi.com/authors/rights for further information.
- Section 3: now section 2. It is better if the authors consider to rename the section as “Botanical Description, Taxonomy and Geographical distribution of Picrasma quassioides”.
- Subsection 2.3: Please, merge this section with that concerning the botanical description, and rename the title as following “Botanical Description and Taxonomy”.
- New subsection: I would insert an additional paragraph on the commercial importance of this plant.
- I think that figures 4-6 should be included in the manuscript, and immediately after being mentioned in the text.
- Section 3: rename as “Phytochemicals in Picrasma quassioides plants”
- Figure 2: according to me it is too big. I would advice the authors to divide the figure into other smaller figures which can be included separately on one-single page. For example, the authors could divide the figures according to family classes. (Fig. 2: Chemical structures of Quassinoid compounds identified in Stem, Wood bark and Leaves of Picrasma quassioides plants; Fig. 3: Chemical structures of β-carboline and Canthinone compounds identified by Stem, Wood and Bark of Picrasma quassioides plants; etc…). The authors should modify the relative captions.
Author Response
Reviewer 2
The manuscript entitled “Review of Phytochemistry, Traditional Use and Pharmacological Activity of Picrasma quassioides” deals with the chemical characterization of Picrasma quassioides, a traditional herbal medicine used in folk and traditional medicine with the aim to treat different kinds of diseases. In addition to the phytochemical composition of this plant, the authors analyzed also its main pharmacological activities, linked to the presence of different classes of bioactive compounds.
The manuscript is generally clear and well written, apart from some minor errors (i.e. the authors do not always quote the Latin name of the plant in italics). It describes a thorough and well-described report, which summarizes the potential of Picrasma quassioides not only in modern medicine, but also in the dietary-supplement field.
The review is quite interesting, however some revisions are necessary.
Response: Thank you very much for the constructive comments. We really appreciate it. All comments have been responded except for removing the methods and splitting the Figure 2. I hope reviewer could agree with our revision.
- Title: I would remove "Review of" at the beginning of the title. If the authors want to specify the typology of the manuscript directly in the title, they could modify it as follows: “Phytochemistry, Traditional Use and Pharmacological Activity of Picrasma quassioides: a critical review” or “Picrasma quassioides: a review on its Phytochemistry, Traditional Use and Pharmacological Activity” or otherwise just leave “Phytochemistry, Traditional Use and Pharmacological Activity of Picrasma quassioides”
- Response: The title has been revised.
- Abstract: in addition to minor error (i.e. Simaroubaceae -> Simaroubaceae; Picrasma quassioides -> Picrasma quassioides, etc..) which are present throughout the manuscript, I think that different parts of the abstract are unnecessary. For example, at line 21 – 23 the authors simply described what a review consists of. I believe this part is unnecessary, and should be removed. For the same reason, remove at Line 23 “As a result”.
Response: Picrasmas quassioides dan Simaroubaceace have been italicised
- Keywords: also here Simaroubaceae should be written in italics. The authors could add other keywords concerning the main families of chemical classes reported and discussed in this work.
- Response: Done
- Introduction: this introduction is too short, and above all it does not address important issues. Some information regarding the economic importance and impact of this plant, should also be added in this section.
Response: There is limited information regarding the economic importance of Picrasma quassioides, because It is harvested from the wild for local use as a food, medicine
- Methods: this paragraph and the related information are not necessary. Please, remove it.
- Response: This method is essential on how the study has been conducted. In literature review, not much methods can be described, but by providing how the information has been gathered explained the content of the study.
- Figure 1 and Figure 2 are not cited in the text.
- Response: Figure 1 and 2 have been mentioned in the text.
- Line 107: it is not enough to write “table below”. Please insert the number of the table.
- Response: No of Table has been included.
- Figure 1: please remove the brackets () from the panel title, and place the CAPITAL letter in the upper right corner of corresponding panel. Modify the caption as follows: “Figure 1: The close-up image of Picrasma quassioides(D. Don) Bennett tree: (A) leaves and fruits; (B) wood bark [11]”. Moreover, I want specify that the authors need the permission to publish photographs for which they do not hold copyright. Please, see https://www.mdpi.com/authors/rights for further information.
Response: the brackets have been removed and the use capital letter. We got permission from the website.
- Section 3: now section 2. It is better if the authors consider to rename the section as “Botanical Description, Taxonomy and Geographical distribution of Picrasma quassioides”.
- Response: The section has been renamed.
- Subsection 2.3: Please, merge this section with that concerning the botanical description, and rename the title as following “Botanical Description and Taxonomy”.
- Response: The section has been merged.
- New subsection: I would insert an additional paragraph on the commercial importance of this plant.
- Response: There is no details concerning the commercial value of the plant to our best knowledge
- I think that figures 4-6 should be included in the manuscript, and immediately after being mentioned in the text.
- Response: The figures have been brought in to the text.
- Section 3: rename as “Phytochemicals in Picrasma quassioidesplants”
- Response: The section has been renamed
- Figure 2: according to me it is too big. I would advice the authors to divide the figure into other smaller figures which can be included separately on one-single page. For example, the authors could divide the figures according to family classes. (Fig. 2: Chemical structures of Quassinoid compounds identified in Stem, Wood bark and Leaves of Picrasma quassioides plants; Fig. 3: Chemical structures of β-carboline and Canthinone compounds identified by Stem, Wood and Bark of Picrasma quassioides plants; etc…). The authors should modify the relative captions.
- Response: Thank you for your concerns regarding the size, but this matter could be managed type setting team from the journal. The description of phytochemicals classes has been grouped in Table 1 and readers just could refer to the compound number to refer.
